# An Optimal Routing Algorithm for Unmanned Aerial Vehicles

**DOI:** 10.3390/s21041219

**Published:** 2021-02-09

**Authors:** Sooyeon Kim, Jae Hyun Kwak, Byoungryul Oh, Da-Han Lee, Duehee Lee

**Affiliations:** 1Department of Electric and Electrical Engineering, Konkuk University, Seoul 05029, Korea; sn8810@konkuk.ac.kr (S.K.); qkdlzldxn@konkuk.ac.kr (B.O.); timeflaw@konkuk.ac.kr (D.-H.L.); 2Department of Electrical and Computer Engineering, University of Rochester, Rochester, NY 60521, USA; jkwak9@u.rochester.edu

**Keywords:** unmanned aerial vehicle, multiple depots vehicle routing problem, subtour elimination, network optimization, mixed integer linear programming

## Abstract

A delivery service using unmanned aerial vehicles (UAVs) has potential as a future business opportunity, due to its speed, safety and low-environmental impact. To operate a UAV delivery network, a management system is required to optimize UAV delivery routes. Therefore, we create a routing algorithm to find optimal round-trip routes for UAVs, which deliver goods from depots to customers. Optimal routes per UAV are determined by minimizing delivery distances considering the maximum range and loading capacity of the UAV. In order to accomplish this, we propose an algorithm with four steps. First, we build a virtual network to describe the realistic environment that UAVs would encounter during operation. Second, we determine the optimal number of in-service UAVs per depot. Third, we eliminate subtours, which are infeasible routes, using flow variables part of the constraints. Fourth, we allocate UAVs to customers minimizing delivery distances from depots to customers. In this process, we allow multiple UAVs to deliver goods to one customer at the same time. Finally, we verify that our algorithm can determine the number of UAVs in service per depot, round-trip routes for UAVs, and allocate UAVs to customers to deliver at the minimum cost.

## 1. Introduction

Unmanned aerial vehicles (UAVs), also known as drones, are very common in today’s society. They are used for pleasure, photography, filmography, and surveillance, but they can also have industrial applications. Amazon, the world’s largest e-commerce company, launched Prime Air [1], a future delivery system using UAVs designed to rapidly deliver goods to customers. At the same time, the COVID-19 pandemic has led to the need to limit physical contact between customers and delivery workers [2]. One option is to replace delivery workers with UAVs to deliver goods safely without contact. Therefore, UAV delivery systems like Prime Air are a prospective business, in that they deliver goods as rapidly and safely as delivery workers can [3].

With the growth of UAV delivery business, delivery management systems will likewise increase [4]. Management systems should form delivery plans that satisfy all customers using UAVs for timely deliveries. Particularly, software that defines UAV routes to minimize travel costs by solving vehicle routing problems are needed. Many previous studies have tried to solve vehicle routing problems for UAVs. In Reference [5], Wang et al. determined optimal routes for two trucks and one drone that uses the trucks as depots. However, we build a routing algorithm solely for UAVs, as UAVs have advantages over trucks, such as flexible deployment, excellent cost-benefit ratios, and minimal environmental impact.

In Reference [6], Dorling et al. found routes for UAVs by minimizing moving distances from a single depot to customers considering the battery depth of discharge. They assumed that the UAV travels in Euclidian distances between customers. However, in this setting, it is hard to consider physical obstacles between customers because the UAV altitude cannot be manipulated. In Reference [7], Zhen et al. defined optimal routes and heights of UAVs in a three-dimensional space using vertical indices from a single depot. They can avoid obstacles by determining vertical indices, but they do not consider obstacles between two vertices. Moreover, these two studies solved routing problems for a single-depot system.

However, unique characteristics of UAVs limit them when delivering goods to customers, so a special environment that can test those characteristics has to be described in the network of UAVs. First, UAVs have a shorter range than ground vehicles like trucks or motorcycles. Therefore, a single-depot system is not suitable for UAVs because they cannot deliver goods to customers who are located outside the maximum range. Second, buildings or trees prevent UAVs from traveling in straight lines from the depot to customers [8], so we need intermediate vertices that UAVs do not necessarily visit but pass by on their way to customers [9].

Furthermore, costs are incurred for using additional UAVs. For example, if the total loading capacity of UAVs exceeds the total customer demand, some UAVs should remain in depots. Therefore, we should solve routing problems using the minimum number of UAVs in service. In Reference [10], the number of electric vehicles in service was calculated by considering different variable costs, speeds, type, and battery capacities of vehicles. However, vehicles in service were assumed to depart from a single depot, so the number of vehicles in service registered over multiple depots, could not be determined. In Reference [11], Lahyani et al. increased the number of depots and defined the UAVs in service for each depot, while solving the vehicle routing problem. Lahyani et al. combined three decisions simultaneously: selecting the number of according to type of UAVs, planning vehicle routes, and assigning routes to depots. Therefore, we also minimize the number of UAVs in service at multiple depots, while planning routes for UAVs.

Most routing problems assume that the designed routes are Hamiltonian cycles [12], where a vehicle visits each vertex exactly once before returning to the starting vertex. For example, in Reference [13], Lim et al. assumed vertices of a network are composed of a depot and customers, so a vehicle started from a single depot, and traveled to all vertex to fulfill customer demand. Furthermore, in Reference [14], Mao et al., assumed that the vertices of the network are composed of multiple depots and customers. UAVs from depots delivered goods to customers and returned to their home depots.

However, if there are intermediate vertices in a network, the designed routes are not a Hamiltonian cycle anymore, so subtours may occur in routes, which indicate that UAVs depart from another vertex, or indicate that routes are not connected. To prevent the occurrence of subtours, it is necessary to add extra constraints. In Reference [15], Ramos et al. suggested new flow variables in constraints to eliminate subtours for multiple depot vehicle routing problems. They also verify the effectiveness of constraints in comparison to more conventional constraints available in the literature. We develop the constraints used in Reference [15] to eliminate subtours.

In a network of UAVs belonging to multiple depots located in different regions, we need to allocate UAVs to customers, while minimizing the distance travelled. In Reference [16], the locations of depots were different, so they assigned UAVs to customers based on previously considered distances between customers and depots. However, a solution gained by the method used in Reference [16] might not minimize cost. In Reference [17], Sung et al. also assign UAVs to specific zones that are divided based on deep learning algorithm previously, but the cost might be not minimum. In Reference [5], total delivery costs using only truck, only drone, and both truck and drone were compared. If the truck and drone are used together, customers that trucks deliver to and customers that drones deliver to are determined by cost minimization.

However, there was only a single depot in the network, so it is difficult to optimally assign UAVs to customers considering the location of depots. Furthermore, two UAVs can deliver to one customer simultaneously, as represented in Reference [18], considering various allocations of the goods to minimize the total cost for delivery. Therefore, we need to solve UAV-customer assignment and routing for UAVs considering that several vehicles can deliver to a single customer.

The papers reviewed for the study are categorized in Table 1. The differences among the research, in terms of the number of vehicles in service, consideration of physical obstacles, the number of depots, existence of intermediate vertices, and assigned UAVs per customers, in Table 1. As the present study deals with a unique combination of the defined categories, it is a valuable addition to the literature on UAV delivery management systems.

In this paper, we introduce a proper network configuration, which has multiple depots and intermediate vertices, to the UAV delivery system. In the network, we determine the number of UAVs in service per depot and solve a routing problem by eliminating subtours for in-service UAVs. While planning UAV routes, we simultaneously assign customers to each UAV, so that the UAV can visit them, and we then determine the quantity that each UAV has to deliver to satisfy the assigned customers. In this process, UAVs satisfy customer demand by considering the loading capacity and maximum range of UAVs.

Our contributions are:We configure a virtual network to implement a realistic UAV delivery system.We optimize the number of UAVs in service per depot to minimize a fixed cost.We find optimal routes for serving UAVs considering the physical limits of UAVs.We eliminate subtours using flow variables in subtour eliminating constraints.We allocate UAVs to customers while solving vehicle routing problems.

This paper is organized as follows. In Section 2, we introduce the configuration of a virtual network composed of depots, customers and intermediate vertices. We also enumerate the objective and constraints for UAV routing problems utilizing flow variables in subtour eliminating constraints. Section 3 verifies the routing results of our formulation. Finally, Section 4 summarizes this paper.

## 2. Unmanned Aerial Vehicles Routing

We build a virtual network to operate UAVs by modeling two realistic environments that UAVs may confront during deliveries. Based on the network, we find optimal round-trip routes of UAVs by minimizing total costs for delivery. In this process, we determine the number of UAVs in service. Furthermore, we eliminate subtours using a flow variable. Moreover, we dispatch UAVs to customers to satisfy demand considering the maximum range and loading capacity of UAVs.

### 2.1. A Virtual Network for UAVs

UAVs must deliver goods to all customers after they depart from their depots without reloading goods. However, a single-depot system is not suitable for UAVs as the maximum range of UAVs can be less than the round-trip distance between the depot and customer. Thus, customers located further than the maximum range from the depot cannot receive goods. Furthermore, UAVs cannot move in straight lines between customers because buildings or trees might block the flight path of UAVs.

Therefore, we build a network having multiple depots organized by region to serve all customers. We also utilize intermediate vertices that UAVs can visit to avoid large obstacles when traveling between customers. In Figure 1, the suggested network configuration is represented based on the 12th AIMMS-MOPTA optimization modeling competition [19].

The network is composed of vertices, which include multiple depots, customers and intermediate vertices, and edges. The blue square markers represent locations of different UAV depots. The yellow triangle markers represent the location of each customer. The gray points represent intermediate vertices for routing UAVs. The gray lines represent edges, and UAVs can only move from vertex to vertex along edges.

The network-related notations are as follows:The vertices set is N={1,2,...,n} denoted by i,j, where i,j∈N, and i≠j.The depots set is P={1,2,...,m} denoted by *p*, where p∈P, P⊂N.The customers set is D={1,2,...,l} denoted by *d*, where d∈D, D⊂N.The set of UAVs belonging to depot *p* is $K_{p}=\left\{1,...,h\right\}$ denoted by $k_{p}$, where $k_{p}\in K_{p}$.The set of all UAVs is *K* denoted by *k*, where $K_{p}\subset K$, K={1,...,s}, k∈K.The set of edges is (i,j)∈{(i,j),...,(s,t)}, where i∈{1,...,n}, j∈{1,...,n}, i≠j.

Based on the network, our objective minimizes total costs for UAVs as follows:(1)$$\min\sum_{k=1}^{s}\sum_{j=1}^{n}\sum_{i=1}^{n}c_{ij}\times x_{ijk}+\sum_{k=1}^{s}\gamma \times \delta_{k},$$
where the integer variable $x_{ijk}$ decides the number of trips between a vertex *i* and a vertex *j* for the *k*th UAV. The variable cost $c_{ij}$ is charged to UAVs moved between *i* and *j*. Furthermore, $\delta_{k}$ is a binary decision variable for the *k*th UAV and works as a service-indicator. If $\delta_{k}$ is 1, then the *k*th UAV is put into use, and, if $\delta_{k}$ is 0, the *k*th UAV is not used. γ is the fixed cost of using a UAV, and *s* is the total number of UAVs registered in depots.

### 2.2. Assigning UAVs in Service

We should use the minimum number of UAVs, as in Reference [20], because the fixed cost γ is charged for a UAV. To minimize the number of UAVs in service, we add the second term in the objective function (Equation 1) using $\delta_{k}$. When $\delta_{k}$ becomes 1, the *k*th UAV is assigned in service, and the fixed cost γ is charged. Finally, the number of UAVs in service per depot is determined by counting the number of assigned UAVs per depot, which is the result obtained by minimizing the objective function (Equation 1).

### 2.3. Eliminating Subtours

We ensure that the routes of UAVs are round-trips by eliminating subtours using the flow variable $q_{ijk}$ in constraints. In Reference [21], conventional constraints for routing UAVs are organized as: (2)$$\begin{array}{cc} \; & \sum_{j=1}^{n}x_{ijk_{p}}=\delta_{k},\;\forall i,\;\forall p,\;i\in P,\;k_{p}\in K_{p}, \end{array}$$(3)$$\begin{array}{cc} \; & \sum_{i=1}^{n}x_{ijk}=\sum_{j=1}^{n}x_{ijk},\;\forall i,\;\forall j,\;\forall k,\;i,j\in N,\;k\in K, \end{array}$$(4)$$\begin{array}{cc} \; & \sum_{k_{p}=1}^{h}\sum_{j=1}^{l}\sum_{i=1}^{n}x_{ijk_{p}}\geq 1,\;\forall p,\;k_{p}\in K_{p}. \end{array}$$

Constraint (Equation 2) enforces that the *k*th UAV in service initiates from its registered depot. Constraint (Equation 3) enforces each vertex is connected to two other vertices, the former vertex and latter vertex, and ensures UAVs finish their routes at their registered depots. Constraint (Equation 4) enforces at least one UAV must visit customers. However, these constraints are not enough to eliminate some subtours. Examples of subtours are represented in Figure 2.

Subtours are infeasible routes. They include examples that UAVs might return to the depot immediately without satisfying customers, as well as that UAVs travel unnecessarily between intermediate vertices and customers. UAVs have to start round trips at depots, so we need restrictions for flows in the network that order UAVs to pass through them while connecting depots and customers.

In order to accomplish this, we need an additional variable associated with edges of the network. In Reference [22], Kara et al. used additional arc-based variables associated with arcs of a network to check the traveling distance of UAVs so that the solution has no subtours. However, the demand for each customer was not considered. Therefore, by modifying the arc-based variables in Reference [22], we suggest flow variable $q_{ijk}$, which indicates the transportation quantity of goods delivered from vertex *i* to *j* by the *k*th UAV. The variable $q_{ijk}$ is solved by a nodal constraint used in a network flow cost minimization problem [23] with an integer variable $x_{ijk}$, which allows $q_{ijk}$.

The sample network with a subtour is described in Figure 3a, and the sample network without a subtour is described in Figure 3b.

Figure 3a represents the flows from the depot (D1) to customers (a,b,c) by minimizing the variable cost and satisfying the nodal constraint [23]. We can see that the nodal constraint ensures flows start from the depot and end at customers. However, if we assume that flows are goods carried by one UAV, we need one continuous route. Furthermore, the UAV must return to the depot after satisfying all customers as in Figure 3b. Therefore, we propose subtour eliminating constraints (Equation 5) and (Equation 8) utilizing an indicator $x_{ijk}$, which indexes flows $q_{ijk}$ according to each UAV and pushes flows to move in a one-way stream, so that the $q_{ijk}$ can represent goods carried by the UAV.

First, we use $q_{ijk}$ in the nodal constraint, which is one of the subtour eliminating constraints, as:(5)$$\sum_{i=1}^{n}q_{ijk}-\sum_{j=1}^{n}q_{ijk}=\pi_{ik},\;\forall i,\forall j,\forall k,\;i,j\in N,k\in K,$$
which states UAVs must start from a depot and satisfy customers. In the nodal constraint (Equation 5), $q_{ijk}$ indicates the quantity of goods delivered by the *k*th UAV from vertex *i* to *j*, and the $\pi_{ik}$ is the total incoming and outgoing quantity to vertex *i* delivered by the *k*th UAV. At a depot *p*, the quantity will be delivered to a customer *d*, so the incoming quantity is 0, and outgoing quantity is positive. Therefore, $\pi_{pk}$ is positive. A quantity is delivered to a customer *d*, so an incoming quantity is positive, and outgoing quantity is 0. Therefore, $\pi_{dk}$ is negative. At the intermediate vertices, the incoming and outgoing quantity is same, so $\pi_{dk}$ is 0. The constraints for ranges of $\pi_{pk}$ and $\pi_{dk}$ are:(6)$$\begin{array}{cc} 0 & \leq \pi_{pk},\;\forall p,\forall k, \end{array}$$
(7)$$\begin{array}{cc} 0 & \geq \pi_{dk},\;\forall d,\forall k, \end{array}$$
and $\pi_{ik}$, which are neither depots nor customers, equal to 0.

Second, we limit $q_{ijk}$ with $x_{ijk}$ as:(8)$$q_{ijk}\leq M\times x_{ijk},\;\forall i,\forall j,\forall k,\;i,j\in N,k\in K,$$
which is the other subtour eliminating constraint. The flow variable $q_{ijk}$ will be 0, if $x_{ijk}$ is 0 because the *k*th UAV does not travel the route between *i* and *j*. The constant *M* is a large number that does not constrain the range of $q_{ijk}$, if $x_{ijk}$ is 1. The term $x_{ijk}$ works as an indicator, which restricts the routes of flow corresponding to the *k*th UAV. As a result, the addition of variable $q_{ijk}$ in constraint (Equation 5), and the indexing of flows by indicator $x_{ijk}$ in constraint (Equation 8) eliminate subtours.

### 2.4. Allocating and Routing UAVs

We allocate UAVs to customers considering the locations of registered depots. In this process, we allow two UAVs to satisfy the demand of one customer at the same time, while finding the routes of UAVs [24]. The routes are determined considering both the maximum range and loading capacity of UAVs.

In the single-depot system in Reference [25], it is not necessary to allocate UAVs to customers because UAVs start from the same depot and visit all customers in the network. However, in a multi-depot system, we have to optimally allocate UAVs to minimize the travel distances from multiple depots to customers. If there is one UAV and two depots D1 and D2 in Figure 4, we must assign the UAV to the optimal depot to minimize delivery cost. Figure 4 shows that allocating the UAV departing from depot D1 costs less than one departing from D2 as D1 is closer to the customers.

Furthermore, two UAVs can deliver goods to a customer together. Therefore, we suggest a decision variable $\pi_{dk}$ that signifies a quantity of goods delivered by a *k*th UAV to a *d*th customer in a constraint:(9)$$\sum_{k=1}^{h}\pi_{dk}=\Pi_{d},\;\forall d.$$

The vector Π is composed of customer demand, so $\Pi_{d}$ is the demand of the *d*th customer. Thus, the sum of $\pi_{dk}$ by *k* should be equal to the *d*th component in the vector Π.

Moreover, we consider the maximum range of UAVs. The maximum range is 150 km following Reference [26], so we limit the total moving distance of UAVs equal to or less than 150 km as:(10)$$\sum_{j=1}^{n}\sum_{i=1}^{n}d_{ij}\times x_{ijk}\leq 150\;\mathrm{km},\;\forall k,$$
where $d_{ij}$ is a distance in km between a vertex *i* and *j*.

We also consider the loading capacity of UAVs as: (11)$$\begin{array}{cc} \; & \pi_{pk}\leq 15\;\mathrm{kg}, \end{array}$$(12)$$\begin{array}{cc} \; & \pi_{pk}=\sum_{d=1}^{l}\left|\pi_{dk}\right|,\;\forall k, \end{array}$$
where $\pi_{pk}$ represents the loading quantity of the *k*th UAV registered in the *p*th depot. We assume that the loading capacity of UAVs is 15 kg following Reference [26], so the loading quantity $\pi_{pk}$ is equal to or less than 15 kg. The value of $\pi_{pk}$ should be equal to the sum of quantity delivered to customers by the *k*th UAV. In this paper, we simply consider the life span of the battery by limiting the range, but non-ideal discharge properties can affect for playing UAVs proposed in References [27,28,29], so it is important to take account of battery’s properties in future work. Lastly, additional constraints for ranges of decision variables are: (13)$$\begin{array}{cc} 0 & \leq q_{ijk},\;\forall i,\forall j,\forall k, \end{array}$$(14)$$\begin{array}{cc} 0 & \leq x_{ijk},\;\forall i,\forall j,\forall k. \end{array}$$

The values of $q_{ijk}$ and $x_{ijk}$ are non-negative.

## 3. Computational Results

We represent the results of round-trip routes for UAVs by simulating the routing algorithm explained in Section 2. First, we estimate the optimal number of UAVs in service per depot. Second, we find optimal round-trip routes for UAVs by eliminating subtours, and verify the result is optimal using sample network. In this process, we also verify that the subtour-eliminating constraints can eliminate subtours. Third, we present delivered quantity $\pi_{dk}$ to the *d*th customer by the *k*th UAV to check that allocated UAVs satisfy all customers. We also verify that make multiple UAVs can deliver to a single customer lower the cost. In this process, we utilize the mixed-integer linear programming solver provided by MATLAB. The primal-dual simplex algorithm is used to solve the mixed-integer problem in this paper [30].

The network has 511 vertices consisting of depots, customers, and intermediate vertices. The network is presented in Reference [19]. The location and demand quantity of customers are given in Reference [19], but we set the number and location of depots, and the number of registered UAVs per depot. The number of depots is 15, and there are five UAVs registered per depot. The number of customers is 75, and the demand quantity is measured from 0 to 10 kg. There are 421 intermediate vertices, excluding the depots and customers.

### 3.1. UAVs in Service by Depot

The optimal numbers of UAVs in service are represented by the translucent blue circle around depots in Figure 5. The size of circle is proportional to the number of UAVs in service. The vertices and location of these depots are represented in blue next to the square markers in Figure 5. All UAVs registered at depots 326 and 476 are used, but none of the UAVs registered at depot 268 are used. At the other depots, either one or two UAVs are used. Depot 476 is isolated from other depots and is the only depot capable of servicing several customers, so all UAVs are in service at this depot. Depot 326 also utilizes all UAVs as it is located in an area of high customer density. However, none of the UAVs registered at depot 268 are used, as using UAVs from depots 294 and 194 is more economical. Depot 194 is located in an area of low customer density, and only a single UAV is in service. Therefore, it can be seen that multiple UAVs are necessary for depots in areas with high customer density and less or none in areas of low customer density.

### 3.2. Round-Trip Routes of UAVs in Service

Optimal round-trip routes of UAVs are figured out by eliminating subtours, while satisfying customer demand for goods. In order to shorten the simulation time, we make a reduced sample network by decreasing the number of vertices and edges for the network in Figure 1. The reduced sample network is only used to verify and visualize the subtour elimination constraints. The reduced network has 3 depots, 15 customers, and 5 UAVs per depot.

Figure 6 shows subtour and round-trip routes. Whereas routings in Figure 6a are unconnected, routings of UAVs in Figure 6b are interconnected and form round-trips due to the subtour elimination constraint by limiting $q_{ijk}$ based on $x_{ijk}$. From depot 156, three UAVs make trips represented by blue, yellow, and green lines. From depot 56, one UAV makes a trip represented by a purple line, and from depot 39, one UAV makes a trip represented by a red line.

In Table 2, we present the details of vertices passed by a single UAV represented by a red line and diamond markers in Figure 6b. In Table 2, Vertices Number gives the vertices passed by the UAV, Loaded $q_{ijk}$ gives the loaded quantity, and Demand Quantities gives the quantity of goods required by customer 47, 110, and 177. The UAV departs from depot 39 and satisfies demand at customer vertices. At customers 47, 177, and 110, values of $q_{ijk}$ decrease by the quantity of demand at those vertices. For example, previous to customer 47, the $q_{ijk}$ was 15, but it decreases equal to the demand of the customer 47, so, after passing customer 47, the loaded $q_{ijk}$ is 5. After all deliveries, the UAV returns to depot 39. Thus, following the results represented in Figure 6 and Table 2, we can verify that subtours do not occur, and goods are delivered completely.

To verify that the result of the routing algorithm based on $q_{ijk}$ is optimal, we apply the algorithm to another simple network, labeled with the costs per edge as in Figure 7. We can assume that the route 1 → 3 → 5 → 6 → 7 → 1 minimizes cost, while delivering goods to customers 5 and 6 as shown in Figure 7a. Since the total cost imposed per edge increases by 4 if the UAV passes vertices 2 and 4, as shown in Figure 7b, the routes do not include those nodes. Therefore, using $q_{ijk}$, we can ensure UAVs make optimal round-trips and satisfy all customers by eliminating subtours.

### 3.3. UAV Allocation for Delivery

UAVs from each depot are optimally allocated to customers, and deliver the optimal quantity of goods considering the maximum range and loading capacity. In Figure 8, we present optimal round-trips of UAVs separated by color in the virtual network. Because of the maximum range constraint, customers 387, 435, and 437 are satisfied by UAVs from depot 434, and customers 387, 232, 414, and 417 are satisfied by UAVs from depot 405. Furthermore, because of limitation of loading quantity, customer 387 is satisfied by two UAVs from depot 434 and depot 405.

In Table 3, we also present the delivered quantity $\pi_{d,k}$, loading quantity $\pi_{p,k}$, and moving distance of the *k*th UAV to the *d*th customer during the *k*th UAV’s delivery. One UAV from depot 36 and 133 deliver to customer 78 and 171. Two UAVs from depot 100 deliver to customers 22, 45, 72, 78, 87, and 171. Five UAVs from depot 476 deliver to customers 290, 297, 305, 314, 320, 325, 338, 343, 350, 352, 380, 393, 399, and 403. Demands of those customers are satisfied by the UAVs completely, within the loading capacity (15 kg) and maximum range (150 km). Furthermore, two UAVs from different depots deliver to the same customer; for example, customer 78 is satisfied by two UAVs: one from depot 36 and one from depot 100. Customer 171 is also satisfied by two UAVs from depot 100 and 131. Moreover, two UAVs from a same depot deliver to a customer; for example, customers 72, 320, and 380 are delivered to by two UAVs from the same depot.

To verify that allowing two UAVs to share one customer’s demand lowers the total cost, we suppose that only a single UAV can deliver to a single customer and determine the number of UAVs in service and routes for them. We present the resulting routes of UAVs in Figure 9. As UAVs cannot split their goods, more are needed to complete deliveries to all customers. We need 2 more UAVs departing from depots 100 and 133, and one more departing from depots 294, 356, and 434 to satisfy customers compared with the results in Figure 8.

Furthermore, we represent the total cost for UAVs in Table 4 and Table 5, when the fixed cost per UAV is 100, and the variable cost is equal to the moving distances of UAVs. The total number of UAVs serving in Table 4 is 26, but the number of UAVs serving in Table 5 is 30. Thus, 3 more UAVs are used to satisfy the customers, and the fixed cost for UAV use increases. Moreover, the total distance that UAVs cover also increases, so the total cost in Table 5 is about 10 percent more than in Table 4. Therefore, these results show that we optimally allocate UAVs to customers by assigning multiple UAVs to a single customer and find routes by considering the maximum range and loading capacity.

## 4. Conclusions

This paper proposes an algorithm to find optimal routes for UAVs, which deliver goods from multiple depots to customers. While delivering, UAVs take off from their depots to customers as long as their loading capacities and range limits are met. We formulate a routing problem for UAVs on the virtual network, where the customers and their demands for goods are given. In the optimal routing process, we optimize the number of UAVs in service per depot to minimize the total fixed and variable costs by observing service-indicator variables in the objective function so that some UAVs may remain grounded at a depot for an entire day. Furthermore, we find the optimal routes for UAVs by eliminating subtours and by optimally allocating UAVs. Our network includes intermediate vertices that UAVs do not necessarily have to pass. The connection of unnecessary intermediate vertices may cause subtours, which result in infeasible routes. We eliminate subtours by adding variables for flows to edges and by limiting flows. Moreover, we optimally allocate UAVs to customers to minimize delivery distances from depots to customers. We also allow multiple UAVs to satisfy the demand of goods of one customer at the same time.

## Figures and Tables

**Figure 1 sensors-21-01219-f001:**
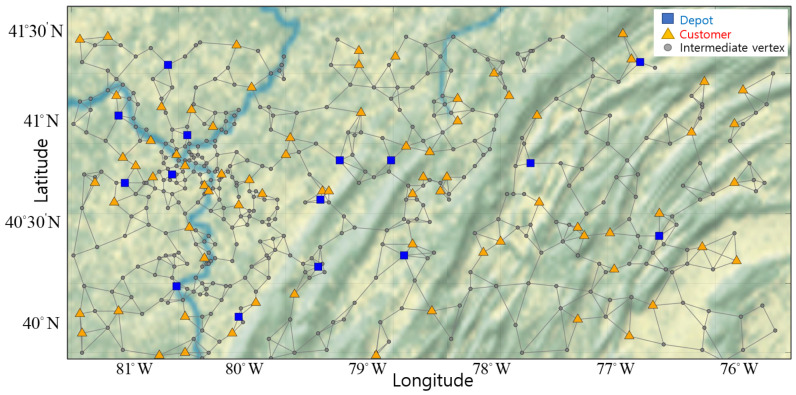
The proposed network configuration with multiple depots, customers, and intermediate vertices.

**Figure 2 sensors-21-01219-f002:**
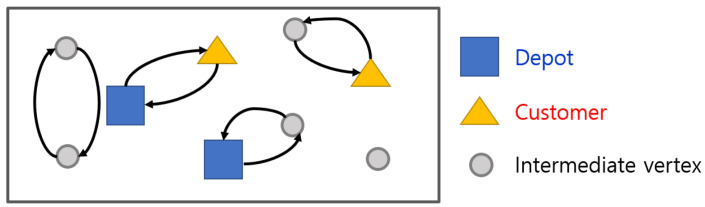
Examples of subtours.

**Figure 3 sensors-21-01219-f003:**
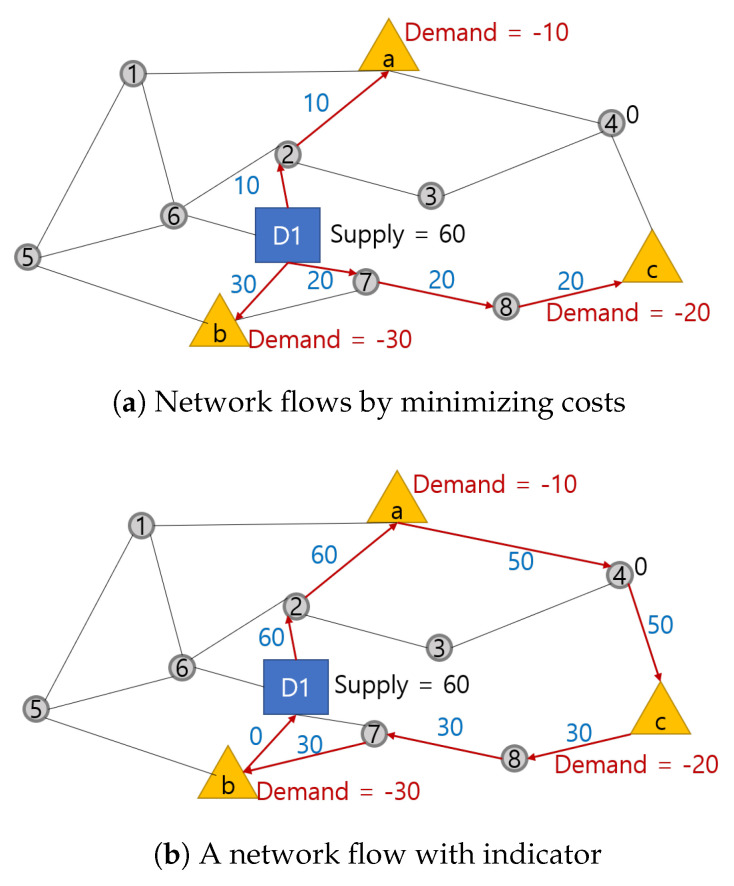
Network flows minimizing costs, with and without an indicator.

**Figure 4 sensors-21-01219-f004:**
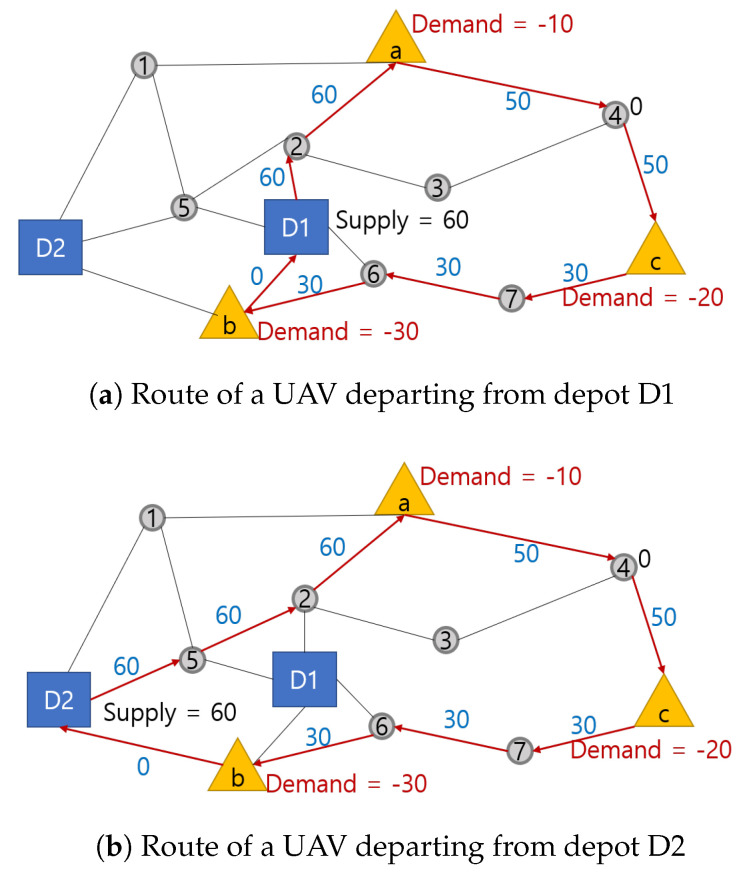
Routes of UAVs from depot D1 and D2.

**Figure 5 sensors-21-01219-f005:**
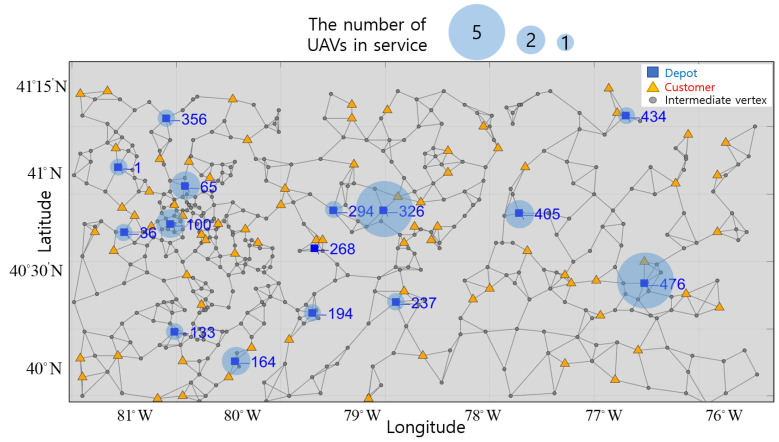
Indices and location of depots in a virtual network.

**Figure 6 sensors-21-01219-f006:**
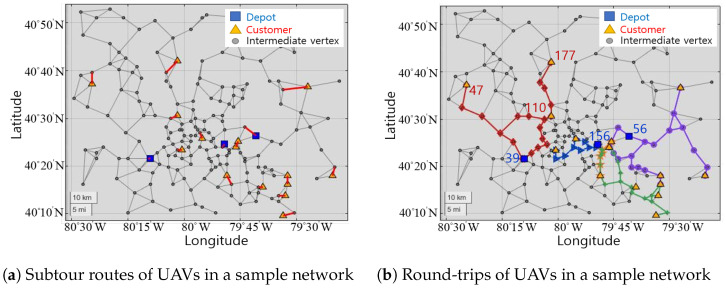
Subtour and round-trip routes of UAVs in a sample network.

**Figure 7 sensors-21-01219-f007:**
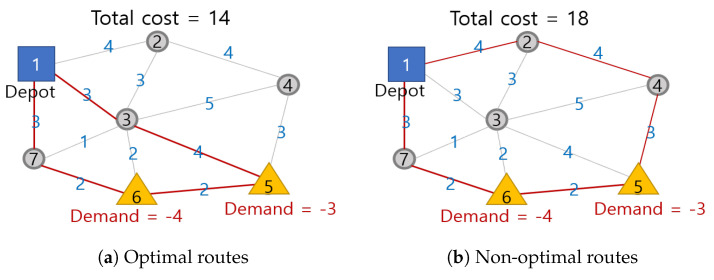
Optimal and non-optimal routes in a simple network.

**Figure 8 sensors-21-01219-f008:**
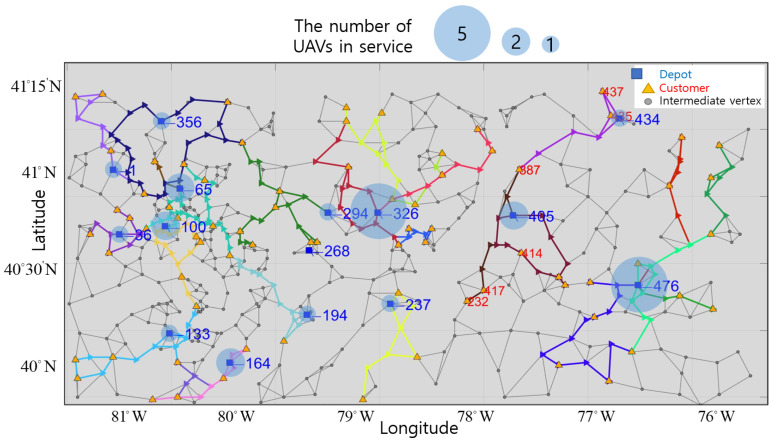
Optimal round-trips of UAVs in the virtual network.

**Figure 9 sensors-21-01219-f009:**
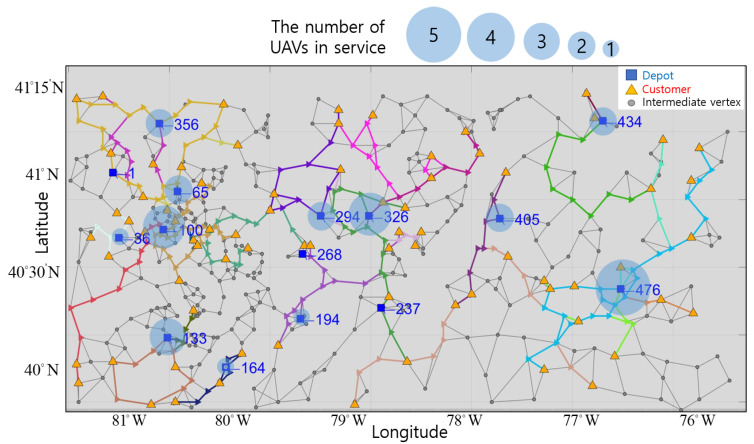
Round-trips of UAVs in the situation that a single UAV can deliver to a customer.

**Table 1 sensors-21-01219-t001:** The research related to the unmanned aerial vehicle (UAV) routing algorithms.

	Vehicles in Service	Physical Obstacles	# of Depots	Intermediate Vertices	Assigned UAVs per Customer
[5]	2-Truck, 1-UAV	x	1-Depot	x	Single UAV
[6]	N-UAV	x	1-Depot	x	Single UAV
[7]	1-UAV	o	1-Depot	x	Single UAV
[8]	N-UAV	x	M-Depot	x	Single UAV
[9]	1-UAV	o	1-Depot	o	Single UAV
[10]	N-UAV	x	M-Depot	x	Single UAV
[11]	N-UAV	x	M-Depot	x	Single UAV
[13]	N-UAV	x	M-Depot	x	Single UAV
[14]	N-UAV	x	1-Depot	x	Single UAV
[15]	N-UAV	x	M-Depot	o	Multiple UAVs
[16]	N-UAV	x	M-Depot	x	Multiple UAVs
[17]	N-UAV	x	M-Depot	x	Single UAV
[18]	N-UAV	x	1-Depot	x	Multiple UAVs
**Present paper**	**N-UAV**	**o**	**M-Depot**	**o**	**Multiple UAVs**

**Table 2 sensors-21-01219-t002:** Paths and delivered quantities of the UAV.

	Network Flow													
Vertices Number	**39**	44	65	15	28	**47**	28	15	65	59	106	83	**110**	116
Loaded $q_{ik}$	15	15	15	15	15	**5**	5	5	5	5	5	5	**2**	2
Demand quantities	0	0	0	0	0	10	0	0	0	0	0	0	3	0
Vertices Number	54	8	**177**	8	54	116	110	83	71	86	101	58	75	**39**
Loaded $q_{ik}$	2	2	**0**	0	0	0	0	0	0	0	0	0	0	0
Demand quantities	0	0	2	0	0	0	0	0	0	0	0	0	0	0

**Table 3 sensors-21-01219-t003:** Quantities delivered by UAVs registered at depots 36, 100, 133, and 476.

Depot Number		36	100		133	476				
Vehicles Number		9	18	19	25	71	72	73	74	75
Loading quantity [kg]		15	15	15	15	15	12	15	15	15
Moving distance [km]		60.5	41.2	62.3	94.0	45.2	109.0	54.4	83.8	83.0
**Customers**	Demands	Delivered quantities								
**22**	1	-	-	1	-	-	-	-	-	-
**45**	9	-	-	9	-	-	-	-	-	-
**72**	10	-	6	4	-	-	-	-	-	-
**78**	7	2	5	-	-	-	-	-	-	-
**87**	4	-	4	-	-	-	-	-	-	-
**171**	2	-	-	1	1	-	-	-	-	-
**290**	8	-	-	-	-	-	8	-	-	-
**297**	2	-	-	-	-	-	-	2	-	-
**305**	1	-	-	-	-	-	1	-	-	-
**314**	7	-	-	-	-	-	-	-	7	-
**320**	9	-	-	-	-	7	2	-	-	-
**325**	2	-	-	-	-	2	-	-	-	-
**338**	8	-	-	-	-	-	-	8	-	-
**343**	2	-	-	-	-	2	-	-	-	-
**350**	4	-	-	-	-	4	-	-	-	-
**352**	3	-	-	-	-	-	-	3	-	-
**380**	9	-	-	-	-	-	1	-	8	-
**393**	10	-	-	-	-	-	-	-	-	10
**399**	5	-	-	-	-	-	-	-	-	5
**403**	2	-	-	-	-	-	-	2	-	-

**Table 4 sensors-21-01219-t004:** Total cost for delivery represented in Figure 8.

Depots Number	UAVs in Service	Moving Distance	Total Cost
1	1	79.05	179.05
36	1	60.52	160.52
65	2	113.57	313.57
100	2	103.53	303.53
133	1	93.98	193.98
164	2	111.16	311.16
194	1	65.77	165.77
237	1	74.1	174.1
268	0	0	0
294	1	129.31	229.31
326	5	375.26	875.26
356	1	106.8	206.8
405	2	154.93	354.93
434	1	73.78	173.78
476	5	451.96	951.96
**Cost**	**2600**	**1993.72**	**4593.72**

**Table 5 sensors-21-01219-t005:** Total cost for delivery represented in Figure 9.

Depots Number	UAVs in Service	Moving Distance	Total Cost
1	0	0	0
36	1	43.81	143.81
65	2	126.31	326.31
100	4	238.45	638.45
133	3	197.76	497.76
164	1	37.16	137.16
194	1	50.84	150.84
237	0	0	0
268	0	0	0
294	2	162.10	362.10
326	4	393.31	793.31
356	2	159.25	359.25
405	2	115.74	315.74
434	2	122.82	322.82
476	5	401.66	901.66
**Cost**	**2900**	**2049.22**	**4949.22**

## Data Availability

The data is available for download on the online page [19].

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
