# Peer review of "An Optimal Routing Algorithm for Unmanned Aerial Vehicles"

_sensors, 2021, doi:10.3390/s21041219_

Round 1

Reviewer 1 Report

The study on the optimal routing of unmanned aerial vehicles (UAVs) is a promising direction in urban logistics. This paper creates a routing algorithm and designs four approaches to find optimal round-trip routes for UAVs. The results verify that the proposed methods are conducive to reducing delivery cost. The authors have clearly presented their ideas. However, there are still some issues that the authors should address further.

  • In the abstract, the authors said “we devise four methods” to optimize the round-trip routes for UAVs. But I can find only three methods from the abstract. Please add the description of the fourth method into the abstract. Furthermore, did you develop four methods or develop one method with four stages?
  • Please review the related works of the four methods in the literature.
  • In line 186, equation (13), I found . It indicates a quantity of goods delivered by the k-th UAV to the d-th customer should be non-positive. I think the quantity should be non-negative. Please explain.
  • It is mentioned in the paper that “multiple UAVs are necessary for depots in areas with high customer density and less or none in areas of low customer density. Moreover, it can be determined the number of using the UAV by population density.” I’d like to know how the quantity of UAVs is determined. I suggest that the authors formulate the number of the UAVs used based on population density.
  • The authors said “Customers served by the UAV are located on vertice 47, 177, and 110. And demand quantities are 10, 3, and 2 for each customer respectively from the Table1.” I cannot directly obtain the demand quantities of vertice 47, 177, and 110 from the Table1, and this paper did not show the related solution method. Please consider revision.
  • In line 228, why does the route not include nodes 2 and 4? I’m confused. Please explain.

Author Response

We appreciate you for your valuable comments, and we write responses on the file attached below.

Reviewer 2 Report

The authors study a variant of the vehicle routing with drones and multiple depots. The main objective is to minimize the costs of using drones. They propose an algorithm to solve the variant and carry out some computational tests.

There is a wide lack in the state of the art. A scarce number of contributions has been considered for the comparison with this work. When the authors state that considering a single-depot in the VRP is not suitable, they should consider that in several VRP and TSP variants with drones, drones are launched from the vehicles, hence, vehicles represent a sort of “mobile depots”.

The algorithm is very basic and the computational test should be extended.

Unfortunately, in my opinion this contribution is not enough for a publication, since several algorithms and more sophisticated versions of the routing with drones have been addressed in the last years.

It is very important to underline the new features of the problem, and be more precise with the literature.

Recently, several updated survey papers have been published, I suggest:

  • Rojas Viloria, D., Solano-Charris, E.L., Munoz-Villamizar, A., Montoya-Torres, J.R., 2020. Unmanned aerial vehicles/drones in vehicle routing problems: a literature review. Int. Trans. Oper. Res.
  • Chung, S.H., Bhawesh, S., L., J., 2020. Optimization for drone and drone-truck combined operations: A review of the state of the art and future directions. Comput. Oper. Res. 105004.
  • G Macrina, LDP Pugliese, F Guerriero, G Laporte Drone-aided routing: A literature review Transportation Research Part C 120,102762 2020
  • Thibbotuwawa, A.; Bocewicz, G.; Nielsen, P.; Banaszak, Z. Unmanned Aerial Vehicle Routing Problems: A Literature Review.  Sci.202010, 4504.

Author Response

(The authors gave the same response as above.)

Reviewer 3 Report

The article optimizes the number of UAVs in-service per depot to minimize the total fixed and variable costs. The introduction to the problem is clear, and the simulations are described comprehensively. 

The only concern from my knowledge is that the authors may be ignoring the battery discharge properties. The high power lithium-ion battery is the most popular power source installed in the drone, while it has several non-ideal discharge properties. The authors focus on the distance. However, the drone's power consumption also plays a critical role in UAV routing problems. 

There are three papers focus on the power consumption of drone when solving the UAV routing issues. I suggest the authors can have a check to strengthen this work. 

"A case for a battery-aware model of drone energy consumption."

"Battery-aware operation range estimation for terrestrial and aerial electric vehicles."

"Energy-Efficient Coordinated Electric Truck-Drone Hybrid Delivery Service Planning."

Line 74 has a typographical error.

Author Response

(The authors gave the same response as above.)

Reviewer 4 Report

This paper provides An Optimal Routing Algorithm for Unmanned Aerial Vehicles.

  1. The ABSTRACT needs to be re-written. What is the background, motivation to do this study? What is the problem statement, what is the contribution? All these things should be briefly described in the abstract.
  2. Unnecessary spacing between lines 74 and 75.
  3. The authors need to provide a table containing the summary of the literature, highlighting the strengths and weaknesses of the previous studies.
  4. The simulation software name and simulation parameters need to be provided.
  5. Equations have been given without citing the source. Similar models are available in other papers as well.
  6. The results presented are based on software simulations only. Can the authors provide experimental results?
  7. The authors fail to provide a comparison of their work with other published papers. How the proposed method is better than others? A comparison table should be provided to justify the contribution.

Author Response

(The authors gave the same response as above.)

Reviewer 5 Report

I) This is not a new problem (VRP) but the issue is still valid. 

II) The article not inluded informations about mothods of the optimization eg. simplex  (the mathematical experiments).

III) The authors don't apply the Sensosrs Microsoft Word template (2021 version): https://www.mdpi.com/files/word-templates/sensors-template.dot or LaTeX template (https://www.mdpi.com/authors/latex) to manuscript preparation. 

Author Response

(The authors gave the same response as above.)

Round 2

Reviewer 1 Report

The authors have addressed my concerns. The paper is acceptable in the current version.

Reviewer 2 Report

The authors improved their article, addressing some critical issues.

Reviewer 4 Report

Thank you for addressing the comments. The revised version of the paper looks good.